# Kidney Transplantation and Diagnostic Imaging: The Early Days and Future Advancements of Transplant Surgery

**DOI:** 10.3390/diagnostics11010047

**Published:** 2020-12-30

**Authors:** Stan Benjamens, Cyril Moers, Riemer H.J.A. Slart, Robert A. Pol

**Affiliations:** 1Department of Surgery, Division of Transplant Surgery, University of Groningen, University Medical Center Groningen, 9713 GZ Groningen, The Netherlands; c.moers@umcg.nl (C.M.); r.pol@umcg.nl (R.A.P.); 2Medical Imaging Center, Department of Nuclear Medicine and Molecular Imaging, University of Groningen, University Medical Center Groningen, 9713 GZ Groningen, The Netherlands; r.h.j.a.slart@umcg.nl; 3Department of Biomedical Photonic Imaging, Faculty of Science and Technology, University of Twente, 7522 NB Enschede, The Netherlands

**Keywords:** kidney transplantation, diagnostic imaging, history, x-ray, computed tomography, renal scintigraphy, ultrasound

## Abstract

The first steps for modern organ transplantation were taken by Emerich Ullmann (Vienne, Austria) in 1902, with a dog-to-dog kidney transplant, and ultimate success was achieved by Joseph Murray in 1954, with the Boston twin brothers. In the same time period, the ground-breaking work of Wilhelm C. Röntgen (1895) and Maria Sklodowska-Curie (1903), on X-rays and radioactivity, enabled the introduction of diagnostic imaging. In the years thereafter, kidney transplantation and diagnostic imaging followed a synergistic path for their development, with key discoveries in transplant rejection pathways, immunosuppressive therapies, and the integration of diagnostic imaging in transplant programs. The first image of a transplanted kidney, a urogram with intravenous contrast, was shown to the public in 1956, and the first recommendations for transplantation diagnostic imaging were published in 1958. Transplant surgeons were eager to use innovative diagnostic modalities, with renal scintigraphy in the 1960s, as well as ultrasound and computed tomography in the 1970s. The use of innovative diagnostic modalities has had a great impact on the reduction of post-operative complications in kidney transplantation, making it one of the key factors for successful transplantation. For the new generation of transplant surgeons, the historical alignment between transplant surgery and diagnostic imaging can be a motivator for future innovations.

## 1. Kidney Transplantation: The Early Days

One of the tales of Saint Cosmas and Damian, as compiled in the Legenda Aurea by Jacob de Voragine, the archbishop of Genoa, Italy in 1265, is an example of the fascination of human beings with transplantation. As described by the archbishop, the twin brothers dedicated their lives to medicine and the church, providing care without the desire for rewards. The most appealing story of their legendary tales is The Miracle of the Transplantation of the Black Leg. In one of the multiple accounts of this legendary tale, a caretaker in the Roman church suffers from cancers in his left leg. As the twin brothers are in Rome, they visit this patient by night and amputate the affected leg. Instead of leaving him crippled, the brothers amputate the leg of a recently buried Moor and use this leg for transplantation [1].

Centuries passed between the compilation of this legendary tale and the initial quest for successful kidney transplantation, started by the surgeon Emerich Ullmann in Vienna, Austria, in 1902. By performing a dog-to-dog kidney transplantation, resulting in five days of kidney transplant function, he motivated European surgeon-scientists to move forward in this medical endeavor [2]. In the same year, Alexis Carrel, a genius surgeon and a controversial political figure, published his first article on the vascular anastomosis of the jugular and carotid artery. His pioneering work on blood vessel anastomosis, as well as experimental vessel and organ transplantation, guided by his mentor Mathieu Jaboulay, is still considered the essential surgical innovation for later human-to-human transplantation. For these efforts, he was awarded the Nobel Prize in Physiology or Medicine in 1912, as the youngest physician-scientist to date [3,4,5]. The next gigantic step for organ transplantation was taken by the Ukrainian surgeon Yurii Y. Voronoy in 1933. With his surgical team from Kherson in Ukraine, he performed the first human-to-human deceased donor kidney transplantation. This surgical intervention resulted in hyperacute rejection and the death of the 26-year-old recipient just two days after transplantation, for which the recipient’s history of mercury intoxication, the extremely long warm ischemia time of more than 6 h, and blood group mismatch are considered to be the main causes [6,7]. However, the potential was recognized, and new initiatives and experiments were started. In 1950, Dempster et al. performed successful dog-to-dog transplantations, with the neck as the surgical site. This also resulted in the first available diagnostic image of the graft, an X-ray image using the injection of a 35% pyelosil contrast fluid (Figure 1) [8].

A brief description of the technique used for this experiment is as follows: “The site used has been the neck where the kidney has been interposed on the carotid–jugular circulation. At the mid-point of the carotid artery, its sheath was stripped for some millimeters, a bulldog clamp applied below and a ligature above; finally, the artery was sectioned between by a sharp razor blade. When suturing was employed, the usual Carrel technique followed; continuous over and over sutures were found satisfactory to effect an end-to-end anastomosis”.

The next steps toward successful organ transplantation took more than 15 years. While the disruptive effect of World War II heavily impacted medical sciences, important innovations took place in the field of vascular surgery, going from vascular ligation to arterial repair [9]. Led by the French surgeon Rene Kuss, a series of kidney transplantations were performed with grafts from convicted felons who were executed by guillotine beheading, and thus acted as donors. While the procedures were a surgical success, the lack of knowledge on immunological matching resulted in transplant rejection in the days following transplantation [10]. In this post-war era, new advances came from the physician-scientist and surgeon Richard Lawler from Illinois, United States. Dr. Lawler and his team performed a human-to-human deceased donor kidney transplantation in a 44-year-old female recipient with polycystic kidney disease. This graft functioned for more than nine months after transplantation, which can be considered a successful intervention from a surgical perspective, and was eventually rejected as immunosuppressive medication was not yet available. In these nine months, the first post-transplant diagnostic imaging procedure was performed, being a retrograde pyelography, which confirmed the diagnosis of a partial stricture of the ureter-bladder anastomosis [11,12].

A breakthrough was achieved with the living kidney donation and transplantation from one identical twin to another by surgeon Joseph Murray and colleagues in 1954. In this historic case, diagnostic imaging was also applied, being a urogram with intravenous contrast [13]. In the following years, the Boston team performed another seven living donor kidney transplantations between identical twins. In six out of these seven cases, normal kidney function was achieved after transplantation, whereas in one recipient, surgical failure occurred due to kinking of the renal artery and subsequent thrombosis. The landmark paper of the description of these cases in the Annals of Surgery (1958) can be considered the first guideline for living donor kidney transplantation, with donor and recipient screening recommendations and descriptions of post-transplant diagnostic procedures (Table 1). Pre-transplant donor and recipient imaging included a chest X-ray, a cystogram, and a urogram with intravenous contrast, whereas post-transplant diagnostics relied on retrograde pyelography [14].

With the introduction of immunosuppressive medication in 1962, starting with 6-mercaptopurine and azathioprine, transplantation with an unrelated living or deceased donor kidney became a viable opportunity [15,16]. Together with the ongoing fight against the immunological barriers for organ transplantation, urological complications showed to be a tough surgical hurdle to take [17]. In an elaborate description by surgeon Thomas Starzl, often referred to as “the father of modern transplantation”, urological complications of the first 216 consecutive transplant recipients are outlined [18]. In all these cases, urograms with intravenous iodine contrast were performed in the first days following transplantation, with a repetition every three to six months. In a subgroup of patients, cystograms and retrograde pyelograms were performed to identify or exclude possible complications [19]. The progress made between 1962 and 1970 can be considered groundbreaking, with a one-year graft survival of 67%, 68%, and 92% for the periods 1962–1964, 1964–1966, and 1966-1968, respectively [20]. The role of advancements in immunosuppressive medication and knowledge about human leukocyte antigen (HLA) matching in this progress is beyond dispute, whereas the alignment between transplant surgery and diagnostic imaging is less well known.

## 2. Historical Alignment between Transplant Surgery and Diagnostic Imaging

Kidney transplantation and diagnostic imaging followed a similar timeline for their development. In 1895, the German physicist Wilhelm C. Röntgen, who was first expelled from the Technical School in Utrecht, The Netherlands, produced the first X-ray image [21]. Only one year later, Antoine H. Becquerel, chairman of Physics of the École Polytechnique in Paris, France, revealed the first evidence of radioactivity [22]. Supervised by Professor Becquerel, Maria Sklodowska-Curie started her eventually ground-breaking work on uranium radiation. Only months after the first dog-to-dog kidney transplantation Professor Becquerel, Maria Curie, and her husband (Pierre Curie) received the Nobel Prize in Physics in 1903 [22,23]. In the same era, with the search for applications of X-ray imaging, the first steps for retrograde pyelography were made by Voelcker and von Lichtenberg, and were generally considered an unintentional discovery [24]. The introduction of intravenous urography, taking another 20 years, was enabled by the work of the urologist Moses Swick, and others, on organically bound iodine (Uroselectan) [25,26]. As described in the reports by Murray and Starzl, these diagnostic imaging techniques proved pivotal for both pre-transplant screening and post-transplantation diagnosis of complications.

In the same year as the first human-to-human kidney transplantation (1933), the scientist Irène Joliot-Curie, daughter of Maria and Pierre Curie, showed that radioactive elements can be produced artificially. This was the essential step toward the discovery of Iodine-131 and Technetium-99m, the cornerstones of nuclear medicine [27,28]. Although the development of kidney transplantation stagnated during World War II, the innovation of X-ray imaging and radioactivity skyrocketed. Radiotracers became a serious factor in medical practice in the 1960s, the golden era for kidney transplantation innovations [29,30]. The medical fields of kidney transplantation and nuclear medicine crossed paths in the application of renal scintigraphy (renography). This technique was first performed using Iodine-131-Hippuran and later replaced by Technetium-99m mercaptoacetyltriglycine (MAG3) in 1988, due to better count statistics and a lower radiation dose [31,32]. In 1965, Figueroa and colleagues published their report on the use of Chlormerodrin (Hg-203) after kidney transplantation in The New England Journal of Medicine (Figure 2) [30]. With the use of radioactive Xenon (Xenon-133), intra-operative renal blood flow measurements were tested in 1967, injecting the radioactive substance directly into the renal artery, while recording the activity over the entire kidney allograft [33]. MAG3 renal scintigraphy is still used for post-transplantation evaluation, mainly for the diagnosis of vascular and urological complications [34].

While X-ray imaging was already widely available, as documented with the first transplant diagnostic images in 1956, the introduction of ultrasound and computed tomography (CT) did not enter clinical practice until the 1950s and 1970s, respectively [35,36]. In a report on postoperative ultrasound examinations, Petrek et al. (1976) described a protocol of ultrasound examinations within 24 h after transplantation for 135 consecutive kidney transplant recipients. These ultrasound examinations of the kidney graft and surrounding anatomical structures focused on depicting and assessing graft enlargement, renal pelvis dilatation, and perinephric fluid collections. In these days, determining graft enlargement without signs of renal pelvis dilatation was considered a reliable sign for acute rejection [37]. In a later prospective study, ultrasound examination showed to be a useful adjunct, with an 81% sensitivity for acute rejection, but to date, it cannot replace histological examination [38]. Early post-transplant ultrasound examinations are still common in many transplant centers; however, these examinations are not used for the diagnosis of rejection [39]. One of the primary reasons to perform these examinations soon after transplantation is to detect transplant renal artery stenosis (TRAS). TRAS is an important diagnosis, as it can result in transplant dysfunction. Several studies have reported the accuracy and clinical applicability of ultrasound examinations for this purpose [40,41].

The specific role of CT procedures in kidney transplantation had still to be determined, with the first record of its application in 1977 [42]. Further steps for broader clinical applications took until 1978, with an overview of CT applications for “renal transplant problems” by Kittredge et al., recommending CT-guided intravenous urography as a first-line diagnostic modality to determine the kidney graft’s position and the extent of surrounding fluid collections [43]. In the following decades, CT procedures were primarily used for living donor assessment and post-transplant screening for malignancies [44]. Currently, CT procedures are a key element of pre- and post-transplantation diagnostic imaging, for both living and deceased donations [45].

While the alignment between transplant surgery and diagnostic imaging slowly decreased after the introduction of ultrasound and CT, imaging innovations gained momentum. This included the progress in molecular imaging, with functional magnetic resonance sequences and innovative positron emission tomography tracers, as well as the implementation of artificial intelligence [46,47,48].

## 3. The Lessons of History

The use of innovative diagnostic modalities has had a great impact on the reduction of post-operative complications in kidney transplantation, making it a key factor for successful transplantation. For the new generation of transplant surgeons, these examples of synergistic cross-disciplinary collaborations between transplant surgery and diagnostic imaging can guide young professionals in using and developing new techniques. However, in past decades, the introduction of new diagnostic modalities has been slow, with only limited use of current imaging advancements in and outside of the surgical field [49]. A possible explanation for the apparent lack of intra- and interdisciplinary innovations could be the ongoing process of sub-specialization and training of super-specialists [50]. When reviewing publications from the early days of transplantation, it becomes clear that medical doctors such as Thomas Starzl, John J. Collins, and Richard E. Wilson were not only talented surgeons but also dedicated scientists with research interests ranging from immunology to radiology. When following the first steps of these predecessors, transplant surgeons could lead the way toward the clinical implementation of artificial intelligence, functional magnetic resonance, and three-dimensional imaging in the field of organ transplantation. International transplant societies, with comprehensive clinical databases, can be considered a solid base for research on artificial intelligence and big data [51,52].

The history of transplant surgery is a tale of unsuccessful and successful attempts, as well as pre-clinical studies for which long-term commitments were required. To enable physician-scientists to explore potential areas for innovation, focus and subsequent funding should be directed at long-term achievements and high-risk proposals. The further development of open access scientific publishing structures will be favorable for the rapid and wide dissemination of innovative research findings.

## 4. Next Steps for Kidney Transplantation and Diagnostic Imaging

It is expected that the aging of the population will continue in the coming decades and that the medical fight against comorbidities, such as obesity and diabetes mellitus, will not be over soon. Knowing this, one can expect that cardiovascular disease and vascular calcification will be a growing concern prior to and after kidney transplantation. Identifying patients who are at risk for major adverse events, due to cardiovascular disease and/or a high degree of vascular calcification, will be essential to improve patients’ outcomes after kidney transplantation. Several studies focusing on CT and ultrasound before and after transplantation have confirmed the importance of diagnostic imaging in the screening for vascular calcification [39,45]. Other advancements in the field of diagnostic imaging are expected to provide a more in-depth view of the extent and progression of calcification. For this purpose, research efforts could focus on the application of (18F)-NaF positron emission tomography (PET) imaging of active calcifications in transplant candidates and recipients. In the general population, (18F)-NaF PET procedures have been shown to be useful in predicting acceleration and assessing the vulnerability of vascular calcification [53]. Using this nuclear medicine technique in combination with modalities for anatomical imaging, such as CT and magnetic resonance imaging (MRI), will enable precise localization of areas with active calcifications. Besides the assessment of vascular calcification, there will be an important future role for cardiac evaluation. Diagnostic imaging procedures for non-invasive assessment of cardiac function include CT coronary angiography and myocardial perfusion imaging (MPI). Future studies are required to establish the role of these procedures for cardiovascular risk stratification [54].

Over time, with important immunosuppression innovations, the one-year incidence of acute rejection has gone from nearly 100% to approximately 10% [55]. Still, acute rejection is an important factor associated with inferior long-term transplants and patient survival [56]. Research efforts should go to early and reliable diagnoses of acute rejection episodes to ensure prompt and adequate therapy. With the pitfalls of kidney allograft biopsies, the current gold standard for the diagnoses of acute rejections, the development of a non-invasive diagnostic tool for this diagnosis can be important for both clinical practice and patient comfort [57]. The first steps have been taken in this direction, with the use of [18F]-fluorodeoxyglucose ((18F)-FDG) PET for the diagnosis of subclinical rejection in humans and (18F)-FDG-labeled T lymphocytes for the diagnosis of acute rat kidney allograft rejection [58,59]. A still ongoing, prospective proof-of-concept study is evaluating the applicability and feasibility of visualizing infiltrating T lymphocytes with [18F]-Fluorobenzoyl-Interleukin-2 ((18F)-FB-IL2) PET [60]. It is the first study using this innovative non-invasive diagnostic tool, radio-labeled interleukin-2 PET, for the diagnosis of acute rejection after kidney transplantation.

Several diagnostic modalities have been studied for the evaluation of transplant function before and after transplantation. Renal scintigraphy and ultrasound are currently used for this purpose; however, both techniques lack specificity and sensitivity for changes in transplant function [32,34,61]. To improve the specificity and sensitivity of ultrasound examinations, the application of ultrasound-compatible contrast agents is being studied. These contrast agents can provide a more elaborate insight into functional changes, such as studies by Grabner et al., and anatomical changes, such as those described by Kazmierski et al. [62,63]. Studies have explored the use of MRI techniques, both functional and anatomical, prior to and after kidney transplantation [64,65]. The positive findings in these studies did not yet convince the broader transplant community to implement the use of MRI in daily transplant practices [47]. To provide solid evidence on the relevance and clinical applicability of functional MRI, larger multicenter studies are pivotal. These studies could use the ongoing advancement in the field of artificial intelligence (AI), combining both functional MRI and AI for pre- and post-transplantation evaluation.

For all described techniques, future advancements could be accelerated by the use of AI and large dataset studies [46,66]. For this purpose, collaborations between the fields of kidney transplantation and diagnostic imaging are essential. Also, financial support from the government and non-governmental organizations will be required to enable large (worldwide) multicenter studies. Transplant organizations, such as the Transplantation Societ*y* and the European Society for Organ Transplantation, should aim for partnerships with diagnostic imaging societies, such as the European Society of Radiology.

## 5. Conclusions

The history of kidney transplantation is also a story of bold innovations in diagnostic imaging, in which interdisciplinary collaborations between pioneer physician-scientists are clearly exposed. Applying state-of-the-art techniques to improve surgical outcomes has proved to be essential for medical advancement. The techniques to improve transplant outcomes can originate from various surgical subspecialties, with the work on vascular anastomosis by surgeon Alexis Carrel, for example, and also from non-surgical specialties, for which multiple examples of diagnostic imaging have been provided in this article. Acknowledging these previous achievements and the alignment between transplant surgery and diagnostic imaging can be a motivator for future innovations. This review provides a brief summary of the potential areas for future diagnostic innovations and the clinical necessity for particular innovations. These examples should be an incentive for the new generation of transplant surgeons to work on their leading role in innovative medicine as surgeon-scientists, efforts that warrant generous support by academic policy-makers.

## Figures and Tables

**Figure 1 diagnostics-11-00047-f001:**
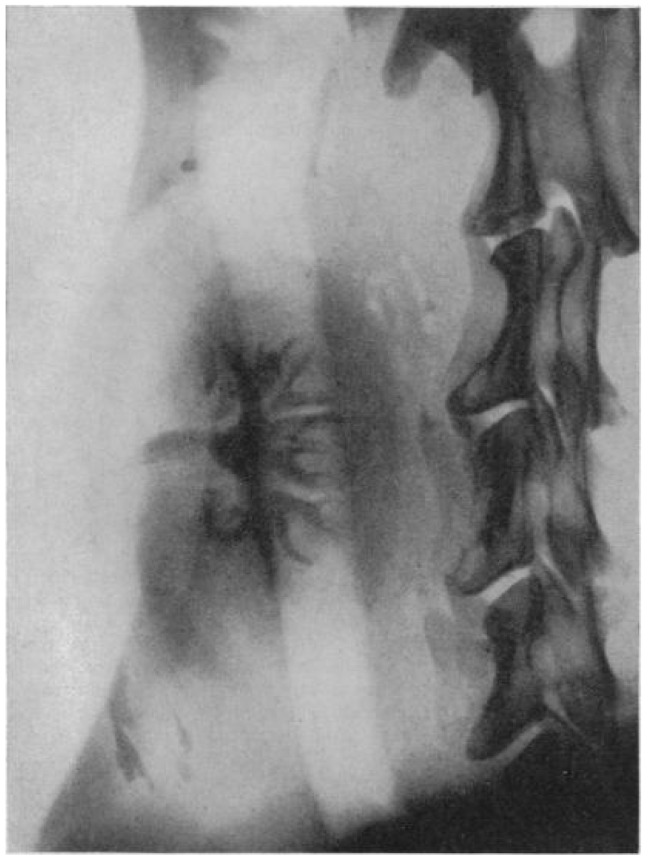
The first available diagnostic image, an X-ray image at 12 min after injection of a 35% pyelosil contrast fluid, of a dog-to-dog transplanted kidney (Dempster, *Ann R Coll Surg Engl.* 1950) [8].

**Figure 2 diagnostics-11-00047-f002:**
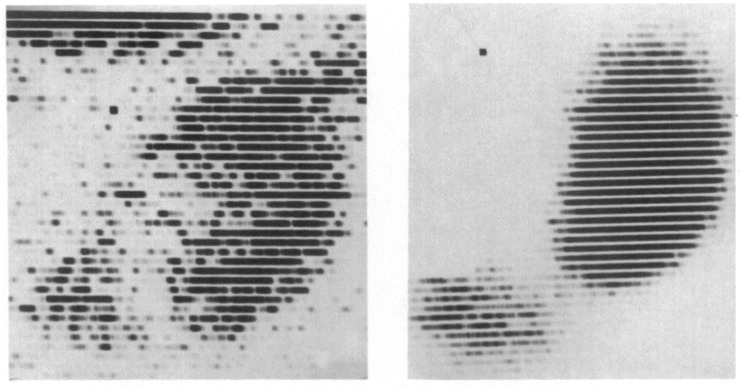
The result of a renal scintigraphy with Chlormerodrin (Hg-203), presented by Figueroa and colleagues in The New England Journal of Medicine. 1965. Left image: “Distinct shadow of the grafted kidney, eleven days after transplantation”; right image: “renal scan thirty-six days after transplantation. Note the great improvement in the renal contour. The shadow in the left lower quadrant is produced by the radioactive urine in the bladder” [30].

**Table 1 diagnostics-11-00047-t001:** Criteria for transplantation, from Murray et al. *Ann Surg*. 1958 [14].

Donor
(1) Two normal kidneys
(2) Normal lower urinary tract
(3) Absence of infection
(4) Sufficient understanding
**Recipient**
(1) Irreversible terminal disease
(2) Normal lower urinary tract
(3) Infection, if present, minimal or controllable
(4) Inactive primary renal disease

## Data Availability

No new data were created or analyzed in this study. Data sharing is not applicable to this article.

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
