# Peer review of "Kidney Transplantation and Diagnostic Imaging: The Early Days and Future Advancements of Transplant Surgery"

_diagnostics, 2020, doi:10.3390/diagnostics11010047_

Round 1

Reviewer 1 Report

The title of the article is "Kidney Transplantation and Diagnostic Imaging: The Early Days of Transplant Surgery".

This article aimed to narrative review the historical alignment between transplant surgery and diagnostic imaging for motivated future innovations in the field.

This is an interesting review that can benefit from more thorough review and appears to be well performed in general and the manuscript is well written. This review may help clinician better assess the historical alignment between transplant surgery and diagnostic imaging. However, the manuscript still could be further improved after some revisions.

  1. Please confirm that all necessary permissions for reproducing or adapting any third-party content in your manuscript have been obtained. Please attach permission certificates.
  2. Please add the historical alignment between ultrasonography and transplant renal artery stenosis including development of the index predicted renal outcome.
  3. Please add more detail about the historical alignment between ultrasonography and transplant renal artery stenosis including development of the index predicted renal outcomes.
  4. It is important that within the manuscript, the authors clarify the importance of this work, the novelty, how it differs from and advances previously published work and how this article can benefit the field and patients in the future etc. Please also add more information from recently published research and offer a more speculative and forward-looking perspective.

Reviewer 2 Report

The authors showed the history, the lessons of history and next steps about kidney transplantation and diagnostic imaging. It sounds interesting. However, this review has some points which need to revise. Please revise the manuscript with these concerns.   Major revisions: 1. Title should be reconsidered. This review showed the history of kidney transplantation and diagnostic imaging, not only the early days of kidney transplant surgery. The title “Kidney Transplantation and Diagnostic Imaging: The Early Days of Transplant Surgerymight be limited to express the whole contents of this review. 2. Table 1 should be organized more clearly because the sentences look irregular shape in the middle of this table, and difficult to understand at a glance. It could be better to separate contents about donor and recipientside by side. Besides, considering recipient (1) and (4) sentences are mentioned about the primary disease, it might be better to put these sentences together.   Minor revisions:

1. Please revise the authors and affiliations. There are no authors whose affiliation is “3: Department of Biomedical Photonic Imaging, Faculty of Science and Technology, University of Twente, 7522 NB Enschede, The Netherlands.

2. In abstract, the description of the first sentence is difficult to understand. Please revise the sentence.

3. Please correct the word “44-years-old” → “44-year-old (line 6, page 3). Also, please correct the word “20-years” → “20 years (line 11, page 3).

4. Please correct the phraseOnly a year later → “Only oneyear later” (line 3, page 4).

5. About acknowledgements, please correct the sentence about Figure 2 (originate from originates from).

Round 2

Reviewer 2 Report

You revered according to my suggestion.